# Photoluminescent Coatings on Zinc Alloy Prepared by Plasma Electrolytic Oxidation in Aluminate Electrolyte

Hanna Maltanava [1,2,*], Stevan Stojadinovic [3], Rastko Vasilic [3,*], Sergey Karpushenkov [4], Nikita Belko [2], Michael Samtsov [2] and Sergey Poznyak [1]

[1] Research Institute for Physical Chemical Problems, Belarusian State University, Leningradskaya Str. 14, 220006 Minsk, Belarus; poznyak@bsu.by

[2] A.N. Sevchenko Institute of Applied Physical Problems, Belarusian State University, Kurchatova Str. 7, 220045 Minsk, Belarus; belkonv@bsu.by (N.B.); samtsov@bsu.by (M.S.)

[3] Faculty of Physics, University of Belgrade, Studentski Trg 12-16, 11000 Belgrade, Serbia; sstevan@ff.bg.ac.rs

[4] Faculty of Chemistry, Belarusian State University, Nezavisimosti Ave. 4, 220030 Minsk, Belarus; karpushenkov@bsu.by

[*] Correspondence: maltanava@bsu.by (H.M.); rastko.vasilic@ff.bg.ac.rs (R.V.)

**Abstract:** Thick $ZnO/ZnAl_2O_4$ coatings were synthesized on zinc alloy Z1 substrates through plasma electrolytic oxidation (PEO) for different anodization times. The prepared coatings were characterized by scanning SEM, XRD, diffuse reflectance and photoluminescence spectroscopy in order to establish the relationship between their structural and optical properties and PEO processing parameters. Under different PEO processing conditions (anodization time—1–10 min and applied voltage—370 and 450 V) ceramic coatings with a mean thickness of 2–12 μm were prepared. XRD analysis explored the coating structure composed of zinc oxide (wurtzite) and zinc aluminate spinel. The content of $ZnAl_2O_4$ in the coatings grows with increasing the applied voltage and anodization time. Photoluminescence (PL) measurements showed that the PEO coatings have several bands in the visible and near-infrared regions associated with their composite structure. The PL spectra significantly depend on the PEO processing parameters due to varying ZnO and $ZnAl_2O_4$ content in the coatings. The insight in the relationship between the $ZnAl_2O_4$ structure and the photoluminescent properties of $ZnO/ZnAl_2O_4$ coatings has been provided using the combination of XRD and luminescence spectroscopy.

**Keywords:** zinc alloy; plasma electrolytic oxidation; ZnO; $ZnAl_2O_4$; photoluminescence





## 1. Introduction

Plasma electrolytic oxidation (PEO) is a powerful tool to create functional layers on the surface of active metals such as aluminium, magnesium, titanium and their alloys [1,2]. This method is based on the material treatment with shortly living (in milliseconds range) discharges in environmentally friendly electrolytes. These discharges are responsible for the conversion of components from substrate and electrolyte into well adherent ceramic-like layers on metal surfaces [3].

At the beginning of PEO treatment, anodic oxide films are typically formed using direct current DC polarization of a metal electrode under potentiostatic, galvanostatic or potentiodynamic control. In most cases, a compact barrier-type film initially grows. As the thickness of oxide film reaches a certain critical value, the film is broken due to impact or tunneling ionization [4–7]. Although the breakdown of anodic layers is considered to be harmful, since it leads to the degradation of their dielectric and protection characteristics, this phenomenon "converts" the process to a next step of metal treatment named "micro-plasma oxidation" or "plasma electrolytic oxidation" [1,2,8–10]. This process operates at potentials above the breakdown voltage of an anodic film and is characterized by luminescence sparks moving over the treated surface. Anodic layers prepared by the

PEO technique are usually rather thick (from fractions of a micron to tens or hundreds of micrometers) [11,12] because a barrier-type film is needed to be formed on a metal surface before the spark discharges can be observed on the electrode surface. Owing to plasma thermochemical interactions in the multiple surface discharges, this method allows obtaining a wide range of the film composition and properties and, therefore, is considered as technologically promising [3]. PEO anodizing has several advantages such as the use of ecological-friendly electrolytes, the preparation of thick coatings without expensive equipment and the absence of special surface treatment before applying the PEO process.

The most of the coatings produced by plasma electrolytic oxidation were investigated on aluminium [12,13], titanium [14], tantalum [15], niobium [16] and zirconium [17], which belongs to the group of valve metals. Nevertheless, in recent years, considerable interest of researchers has been directed to the production of PEO coatings on some non-valve metals, such as magnesium [18], zinc [19,20], iron [21–23], etc. Compared to magnesium, studies on the PEO process on zinc are very limited, although the first patent was published in 1967 [14]. For preparation of PEO coatings on Zn, non-concentrated alkaline solutions were used without any additives or with addition of sodium silicate, sodium aluminate or sodium phosphate [15–18]. Although the possibility of obtaining anodic coatings on zinc in the PEO regime was demonstrated, the various physico-chemical properties of such coatings have not been studied sufficiently.

PEO coatings obtained in aluminate electrolytes can demonstrate improved functional properties due to the formation of $ZnAl_2O_4/ZnO$ heterostructures [24]. Zinc aluminate ($ZnAl_2O_4$) is a spinel type oxide which characterized a wide band gap (~3.8 eV), chemical and thermal stability, low surface acidity, high mechanical resistance, superior optical transmittance and high fluorescence efficiency [25,26]. All these properties make $ZnAl_2O_4$ a suitable material for various applications, such as photoelectronic devices [27], optical coatings [28], electroluminescence displays [29], catalyst and catalytic support [30–33]. Recently, $ZnAl_2O_4$ spinel has given interest as a cathode material in Li-ion batteries [34], UV-emitter [35], supercapacitor [36] and $ZnAl_2O_4@NiFe_2O_4$ composite magnetic sensor [37]. Zinc oxide is an extensively studied semiconductor material that is of interest for photocatalysis and light-emitting diodes [38,39]. $ZnO/ZnAl_2O_4$ heterostructure has given the new possibility for science and industry due to diversity of morphology, composition and long term stability of the composite. Thus, a series of $ZnO/ZnAl_2O_4$ composites have been prepared for photocatalytic degradation of organic dyes [40,41]. However, photoluminescence and optical properties of $ZnO/ZnAl_2O_4$ heterostructures have not been completely understood. It is clearly of relevance to explore such properties of the $ZnAl_2O_4/ZnO$ composites obtained via PEO process since some synergy effect between ZnO and $ZnAl_2O_4$ semiconductors can be anticipated.

The goal of the present work is the preparation of oxide coatings on zinc alloy in alkaline sodium aluminate-containing electrolyte in PEO mode to improve photoluminescence (PL) properties of the metal surface. In addition, the relationship between the microstructure, chemical and phase composition of the PEO coatings on the Zn alloy and their semiconducting and photoluminescence properties was studied.

## 2. Materials and Methods

### 2.1. Materials

For preparation of PEO coatings, zinc alloy Z1 (according to EN988) (VMZinc, Bagnolet, France) with a nominal composition [wt.%]: 0.08 ÷ 1.00% Cu, 0.06 ÷ 0.20% Ti, ≤0.015% Al and Zn balance was used as a substrate. 1.5 mm thick metal foil was cut into 12.5 mm × 20 mm pieces and a small hole was drilled in the upper part of each piece for an electrical contact. The Zn alloy samples were cleaned with ethanol in an ultrasonic bath, washed with deionized water, and then attached to a titanium rod coated with Teflon. The connection spot was insulated with silicone sealant.

An aqueous solution containing 8.2 g/L $NaAlO_2$ (Sigma Aldrich, Louis, MO, USA, technical, anhydrous) and 2 g/L KOH (Sigma Aldrich, Louis, MO, USA, 90%) was used

as the electrolyte. The electrolyte was prepared using deionized water and analytical-grade chemicals.

## 2.2. Setup for PEO Anodization

The power source for PEO anodizing was a home-made DC power supply unit, providing rectified voltage from 0 to 500 V and current up to 3 A. Anodic oxidation of zinc electrodes was carried out in a cylindrical glass cell with a volume of 700 mL. A zinc alloy plate and a 40 mm $\times$ 15 mm stainless steel plate were used as the anode and cathode in the experiments, respectively. During the anodizing process, the electrolyte in the cell was mixed with a magnetic stirrer. The temperature of the electrolyte was maintained in the range of 10–50 °C using a cooling system.

A voltmeter and an amperometer were used to control the output parameters of anodizing. To measure the transient current in the system, a precision resistor (shunt) was connected in series to the circuit and the potential drop on this resistor was recorded by a digital oscilloscope.

## 2.3. Characterization Techniques

The phase composition of the prepared coatings was examined by X-ray diffraction (XRD) method on a PANalytical X'Pert PRO MRD (Multi-Purpose Research Diffractometer, Almelo, The Netherlands) in Bragg-Brentano geometry using CuK$_\alpha$-radiation. Recording speed was 0.4°/min. A JEOL 840 A scanning electron microscope (SEM) (JEOL Ltd., Tokyo, Japan) equipped with an X-ray energy dispersive spectroscopy (EDS) setup was used to characterize the morphology and chemical composition of the formed oxide films.

PL spectral measurements were performed on a Horiba Jobin Yvon Fluorolog FL3-22 spectrofluorimeter (Horiba, Palaiseau, France) at room temperature, with a 450 W Xe lamp as the excitation light source. The obtained spectra were corrected for the spectral response of the measuring system and spectral distribution of the Xe lamp. UV–vis diffuse reflectance spectra (DRS) of the formed coatings were recorded using a UV–vis spectrophotometer (Shimadzu UV-3600, Kyoto, Japan).

Spectral characterization of the sparks appearing during the PEO process was performed using a grating spectrometer with an intensified charge coupled device (ICCD). Optical detection system consisted of a large-aperture achromatic lens, a 0.3 m Czerny-Turner type monochromator (Hilger spectrometer, diffraction grating 1200 grooves/mm, inverse linear dispersion of 2.7 nm/mm in the first diffraction order and wavelength range of 43 nm) (Hilger Crystals, London, UK) and a very sensitive PI-MAX ICCD thermoelectrically cooled camera (−40 °C) manufactured by Princeton Instruments. (Trenton, NJ, USA) Inverse linear dispersion of the optical detection system was 0.07 nm per pixel. The system was used with several grating positions with overlapping wavelength range of 20 nm. Spectra of sparks obtained by this system were recorded using the integration time of 0.1 s. In all experiments the image of the cathode surface was projected with unity magnification to the entrance slit of the spectrometer. The obtained spectra were adjusted to the spectral response of the measuring system.

A low dispersion system fiber optic spectrometer USB4000 UV/VIS (Ocean Optics) (Orlando, FL, USA) was used for the measurements in the spectral range from 400 nm to 800 nm. The spectrometer detector consisted of a 3648-elementar linear CCD array with a diffraction grating of 600 grooves/mm. The light emitted during PEO process was transmitted from Ocean Optics QP400-2-UV/BX on the spectrometer slit and recorded with integration time of 1 s.

## 3. Results and Discussion

### 3.1. Electrochemical Behavior of Zinc Alloy Electrodes at Anodic Polarization in Alkaline Aluminate-Containing Electrolyte

In the present research, a unipolar pulsed DC mode, i.e., under only positive polarization of the metal electrode, for PEO anodization was used. Initially, the voltage on

the electrochemical cell was increased with a rate of 1 V/min until a certain voltage was reached. Then anodization was carried out in the potentiostatic mode. Figure 1 shows typical current density data as a function of time during the PEO anodization of zinc alloy. During linear increase of the voltage to about 370 V, the current density passed through a maximum followed by a sharp drop. At this stage, the appearance of a large number of small microdischarges with white color was observed ("soft sparking" mode). In potentiostatic mode at 370 V, the coating continues to grow and the current drops because of this. The appearance of a current density peak after ~200 s of anodization is associated with an increase in the size of the microdischarges and a decrease of their total number. This phenomenon is ascribed to reduced number of discharging sites through which a higher anodic current is able to pass [42]. After 300 s at 370 V of PEO treatment, the change of color of sparks into yellow ("spark" mode) was observed.

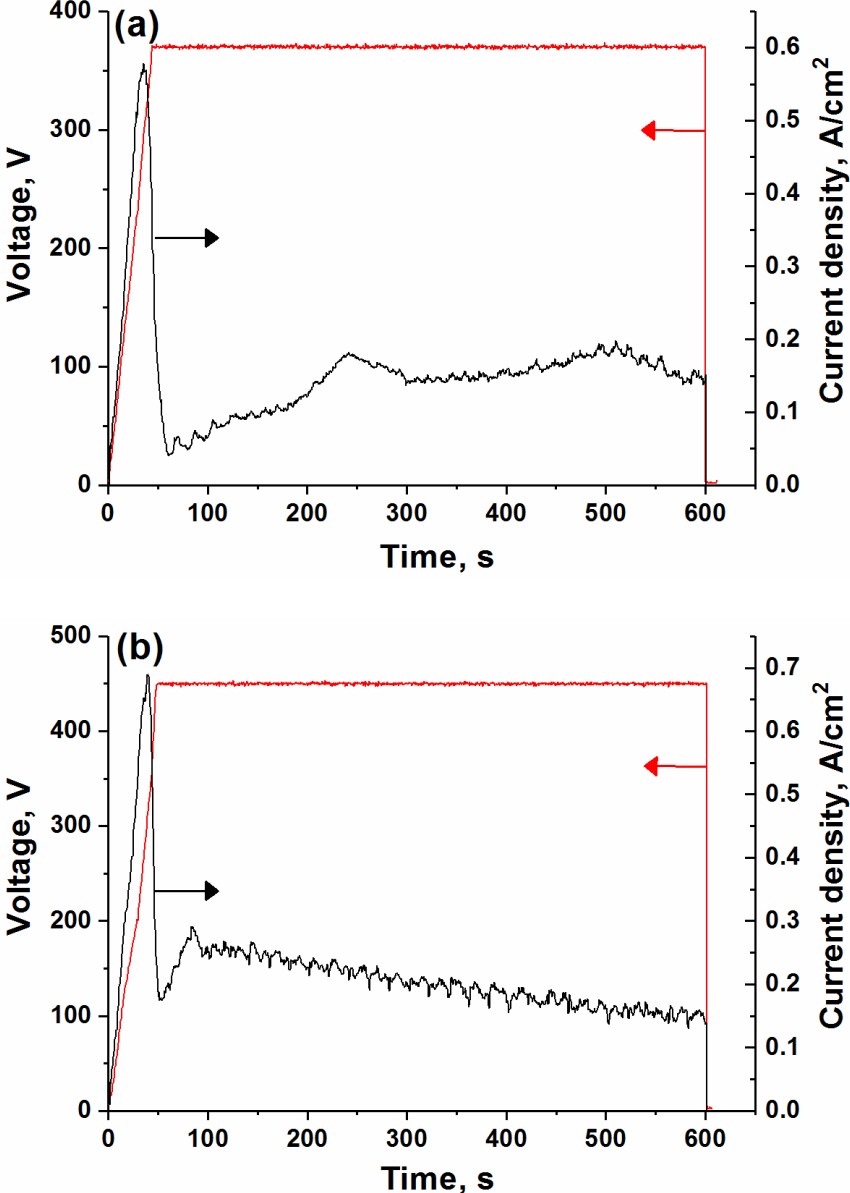

**Figure 1.** Temporal evolution of the current density and peak voltage during anodization of Zn alloy (**a**) peak voltage 370 V; (**b**) peak voltage 450 V.

The increase of voltage to 450 V leads to a larger current density peak associated with more intensive oxygen bubbling. Moreover, first sparks appear after about 3 s of anodizing and rapidly change their color from white to yellow. After ~100 s of anodizing,

the current density gradually decreases with anodizing time, and visual observations reveal the occurrence of fine homogeneously dispersed microarcs. Increasing the voltage above 450 V intensifies the microarcs formation and gas release, which leads to peeling of the oxide layer. Since the PEO coatings prepared at 370 and 450 V are uniform and have rather good adhesion to the Zn substrate (see Appendix A), this peak voltage range was selected for further studies of the coating growth on Zn alloy. The wear behavior of prepared PEO coatings is presented in Appendix A (Figures A1 and A2).

### 3.2. PEO Coating Morphology and Elemental Composition

The typical surface morphologies of two sets of PEO coatings, prepared under 370 and 450 V for different time periods, are presented in the SEM micrographs in Figures 2–5. SEM inspection showed that the prepared samples have the fused surface with randomly distributed microcraters. These microcraters or micropores are characteristic of the PEO coatings and were previously assigned to the gas bubble emission through the molten materials or to the formation of very energetic discharges across the growing film [43,44]. As can be seen from Figures 2–5, the size of the microcraters grows with increasing the voltage and anodization time. Moreover, the coatings grown at higher voltage (450 V) exhibit a rougher surface with some microcracks (Figure 4). These microcracks could be attributed to the thermal stresses during the coating growth as a result of melting and solidification of the ceramic compounds such as zinc oxide.

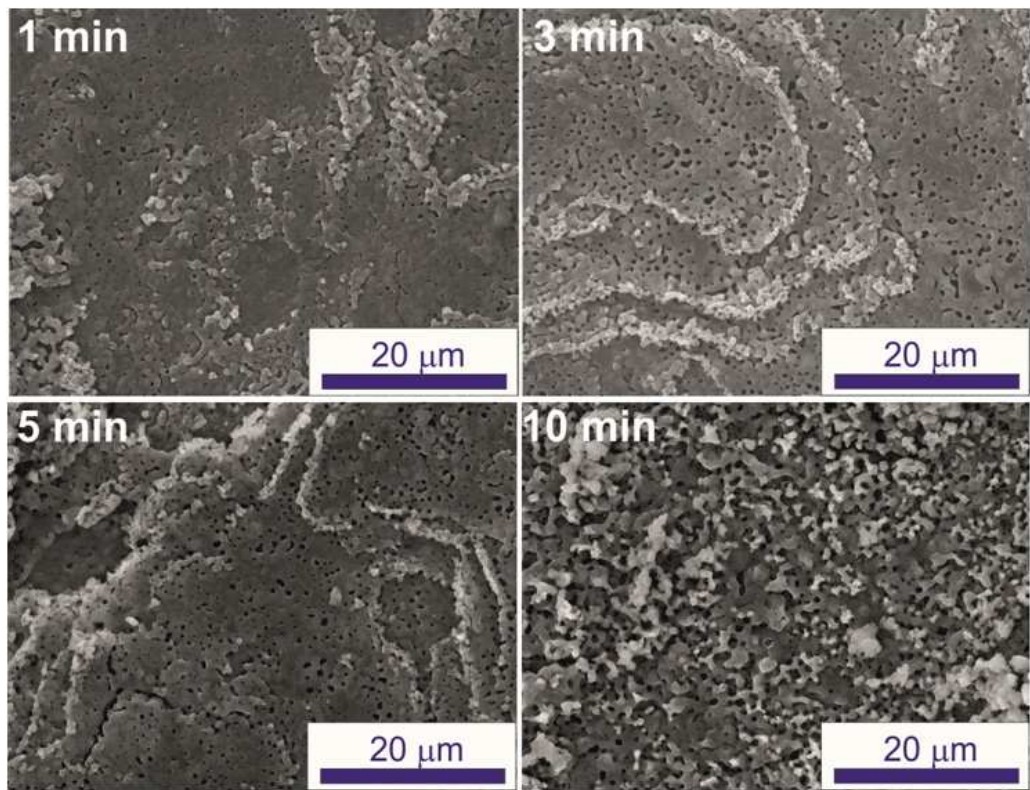

**Figure 2.** SEM micrographs (lower magnification) showing surface morphology of the PEO coatings prepared at 370 V for different anodization time.

The presence of a significant Al content along with Zn and O in the EDS spectra of these coatings indicates that Al is included into the coatings owing to aluminate anions present in the electrolyte. It is important to note that an increase in the processing voltage from 370 to 450 V leads to a rise in the aluminium content and a decrease in the zinc content in the PEO coatings.

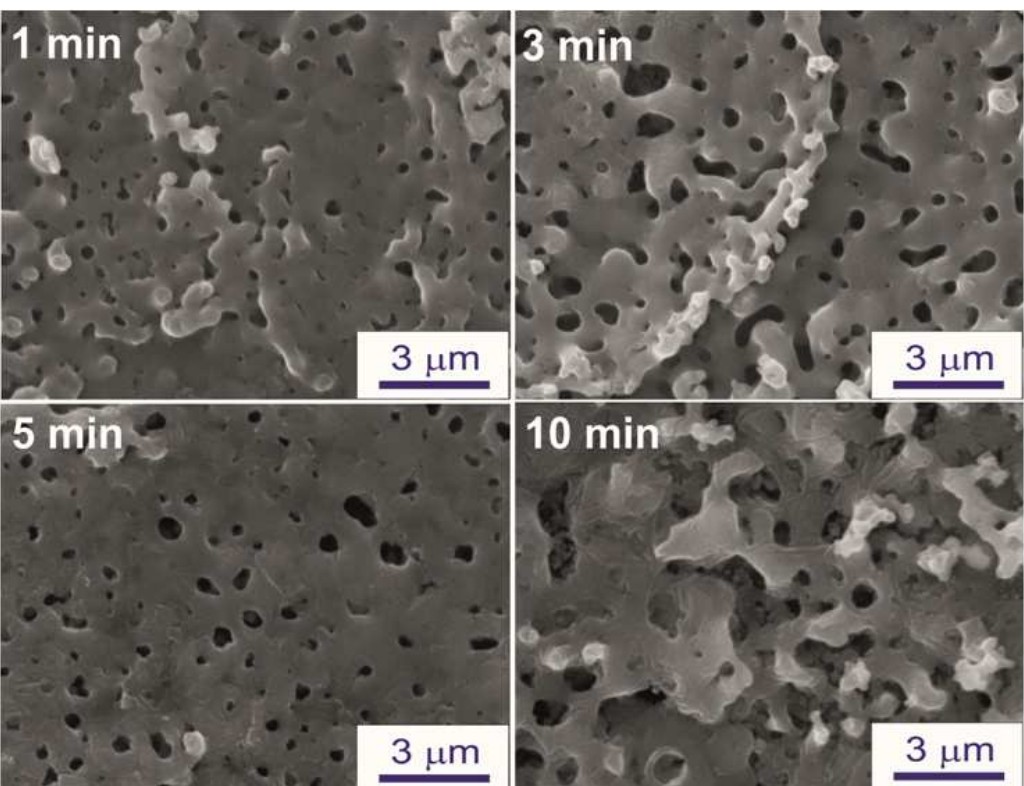

**Figure 3.** SEM micrographs (higher magnification) showing surface morphology of the PEO coatings prepared at 370 V for different anodization time.

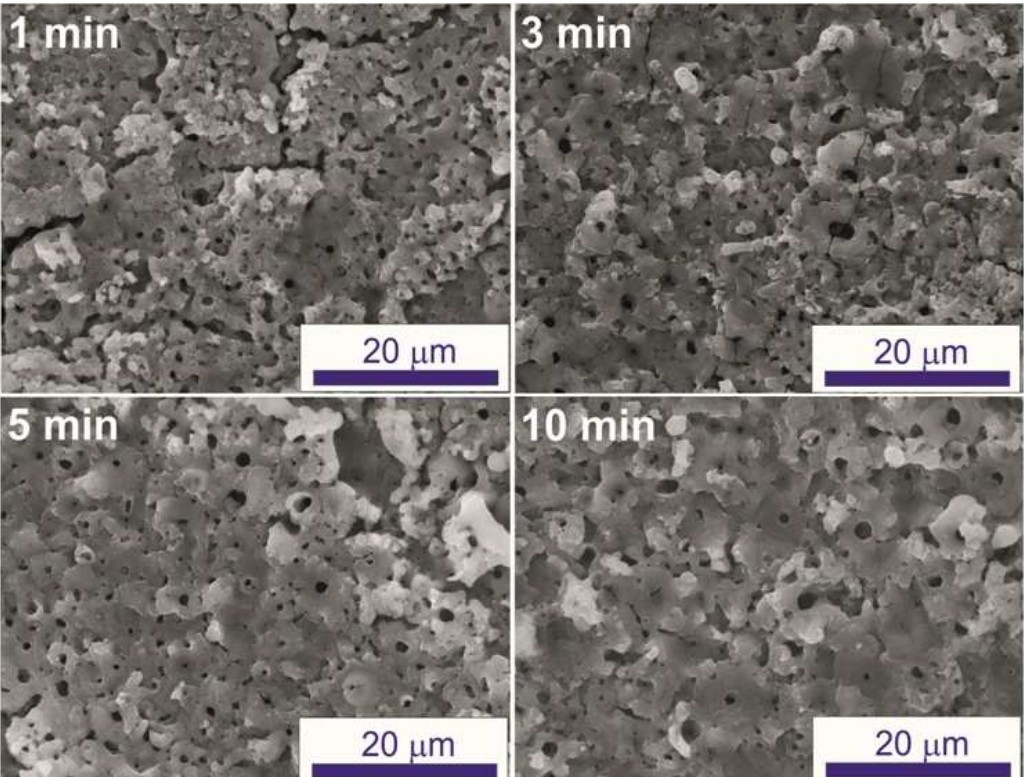

**Figure 4.** SEM micrographs (lower magnification) showing surface morphology of the PEO coatings prepared at 450 V for different anodization time.

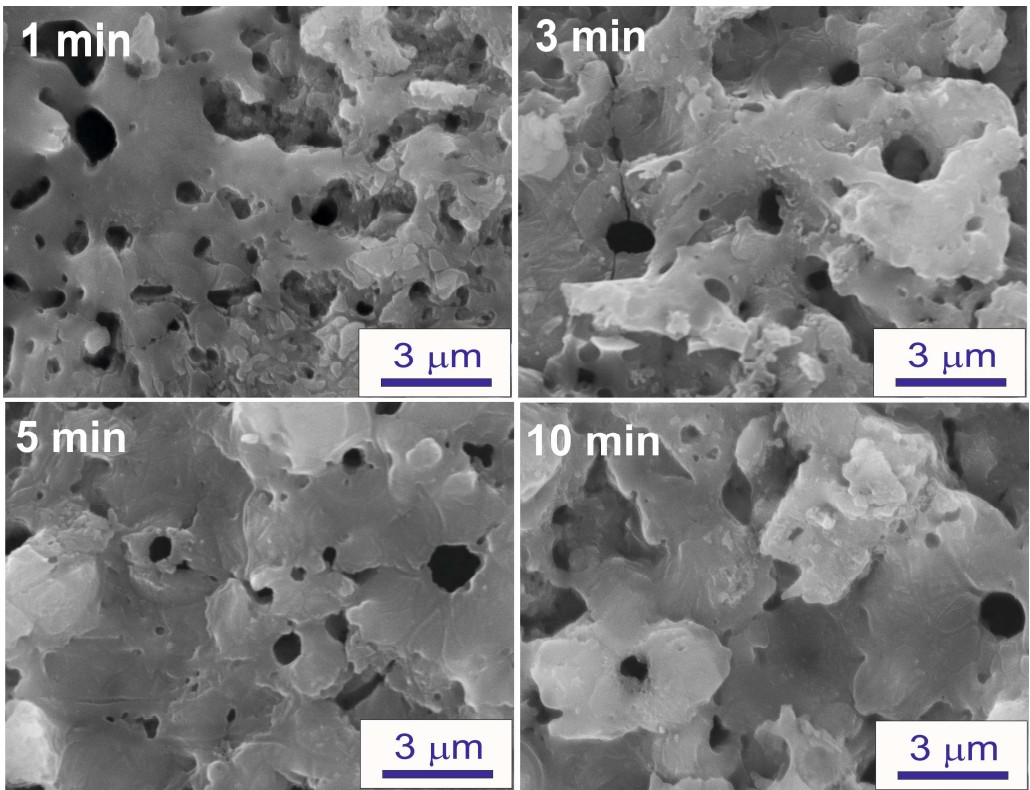

**Figure 5.** SEM micrographs (higher magnification) showing surface morphology of the PEO coatings prepared at 450 V for different anodization time.

Results of the EDS analysis of the PEO coatings on Zn are shown in Table 1. Main elements of the coatings are determined as O, Zn, and Al.

**Table 1.** Chemical composition of the PEO coatings (in at.%).

| Time of the Treatment, min | Applied Voltage: 370 V | | | Applied Voltage: 450 V | | |
|---|---|---|---|---|---|---|
| | **O** | **Al** | **Zn** | **O** | **Al** | **Zn** |
| 1 | 59.62 | 26.30 | 14.09 | 60.66 | 26.40 | 12.93 |
| 3 | 60.72 | 27.62 | 11.61 | 60.97 | 31.05 | 7.98 |
| 5 | 60.71 | 27.13 | 12.16 | 62.90 | 30.61 | 6.49 |
| 10 | 59.74 | 24.09 | 16.17 | 59.93 | 33.04 | 7.03 |

Figures 6 and 7 show SEM micrographs of cross-sections of the PEO coated Zn alloy samples prepared at two different voltages (370 and 450 V) for treatment times of 1 and 10 min. All coating-substrate interfaces have an uneven appearance, which may be the result of melting or dissolution of the substrate during the PEO treatment. The coating thickness after 1 min treatment at 370 V is about 2–4.5 μm at different locations of the cross sections. This thickness increases to 4.5–6.5 μm with a rise in processing time up to 10 min (Figure 6). PEO anodization of Zn electrodes at 450 V leads to an increase in the coating thickness (4–6.5 μm after 1 min treatment and 8–12 μm after 10 min treatment) (Figure 7). The obtained data demonstrate that an increase in the processing time from 1 to 10 min results in only a 2-fold rise in the coating thickness. This fact can be explained by the partial detachment of the coating material due to the abundant gas evolution during the anodizing process.

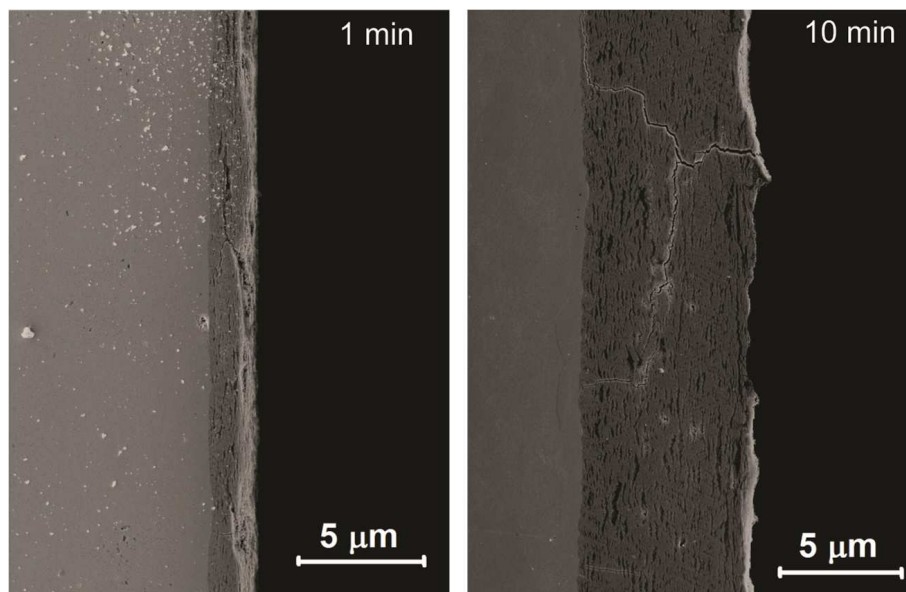

**Figure 6.** SEM micrographs of cross-sections of the PEO coatings grown at 370 V for 1 and 10 min.

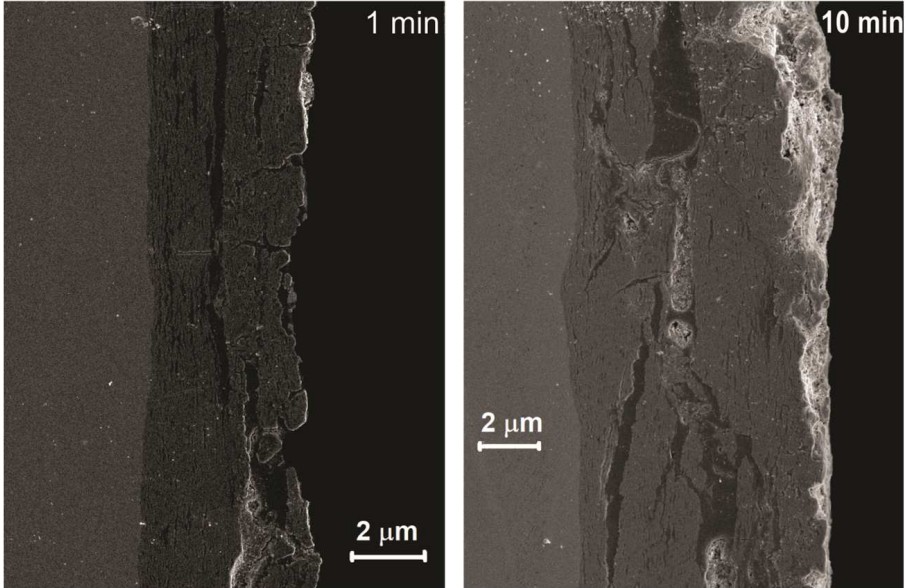

**Figure 7.** SEM micrographs of cross-sections of the PEO coatings grown at 450 V for 1 and 10 min.

The EDS mapping of the cross-sections of the PEO coatings obtained at 370 and 450 V revealed uneven distribution of zinc and aluminium over the depth of the coatings. It has been identified that the incorporation of aluminium as zinc aluminate into the coatings is preceded by the formation of a zinc oxide layer (as will be shown below). The cross-sectional view of the PEO coatings obtained at 370 V demonstrated the deposition of alternating layers of zinc oxide and zinc aluminate as the coating thickness grows (Figure 8a). Such structure can result from the synthesis of $ZnAl_2O_4$ during high energy plasma discharges. At the same time, two regions are observed for coatings obtained at 450 V (Figure 8b). The inner region is composed of ZnO, whereas the outer region is mainly presented by a thick layer of zinc aluminate. The structural features of the samples treated at 450 V can be related to avalanches generated in zinc oxide, which lead to the occurrence of microarcs and contribute to the synthesis of the aluminate layer.

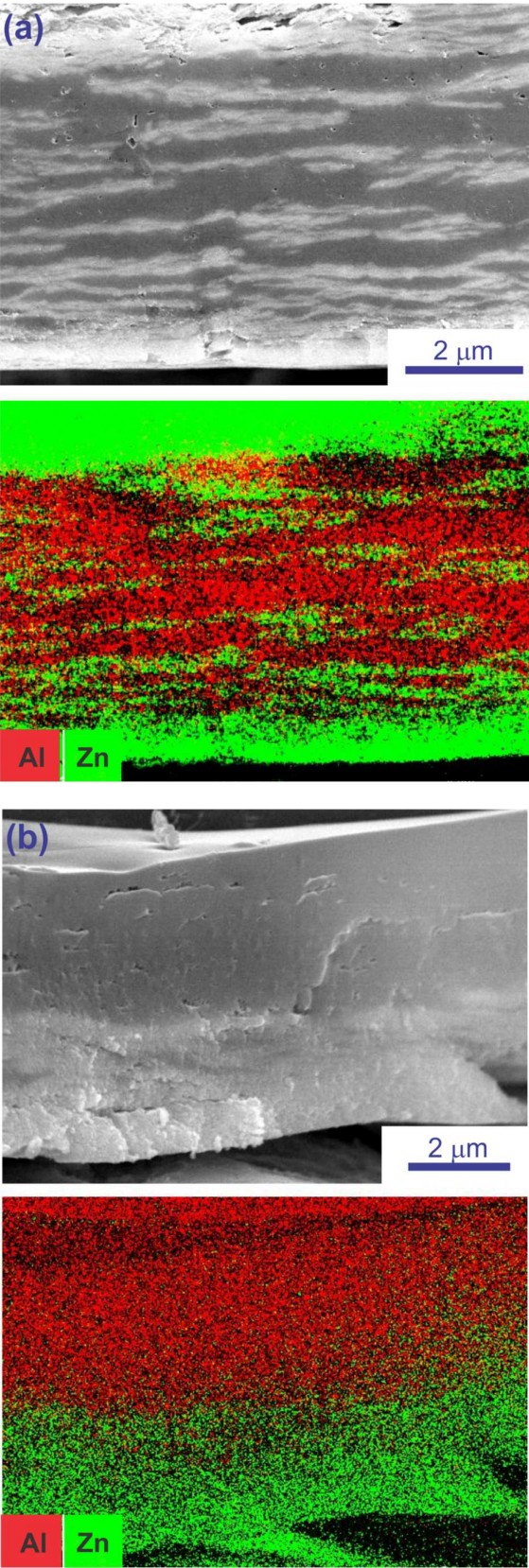

**Figure 8.** SEM micrographs and associated EDS mapping of cross-sections of the PEO coatings grown at 370 V (**a**) and 450 V (**b**).

### 3.3. Crystalline Structure of the PEO Coatings

The XRD patterns of the PEO coatings formed at 370 V for different processing time are shown in Figure 9. Beside the characteristic diffraction peaks of the Zn alloy substrate (due to thin thickness of the coatings), ZnO and $ZnAl_2O_4$ phases were detected in all the X-ray diffractograms. ZnO has wurtzite structure with (100), (002), (101), (110), (103), (200), (112), (201), (004) and (201) diffraction peaks at 31.8, 34.4, 36.3, 56.6, 62.9, 66.4, 68.0, 69.1, 72.6 and 77°, respectively. $ZnAl_2O_4$ exhibits a spinel structure with (220), (311), (400), (422), (511), (440), (620) and (533) diffraction peaks at 31.2, 36.8, 44.8, 55.6, 59.3, 65.2, 74.1 and 77.3°. It is known that aluminates (such as $MgAl_2O_4$ [45]) can be of different stoichiometry. The XRD peaks for $ZnAl_2O_4$ show excellent agreement with the literature data indicating that the stoichiometry for ZnO to $Al_2O_3$ in this phase is close to 1:1. At short processing times, the ZnO phase is the main one, while with an increase in the anodization time, the $ZnAl_2O_4$ phase becomes predominant (Figure 9). When the anodization voltage is increased to 450 V, the relative $ZnAl_2O_4$ content in the coating grows even at short processing times (Figure 10).

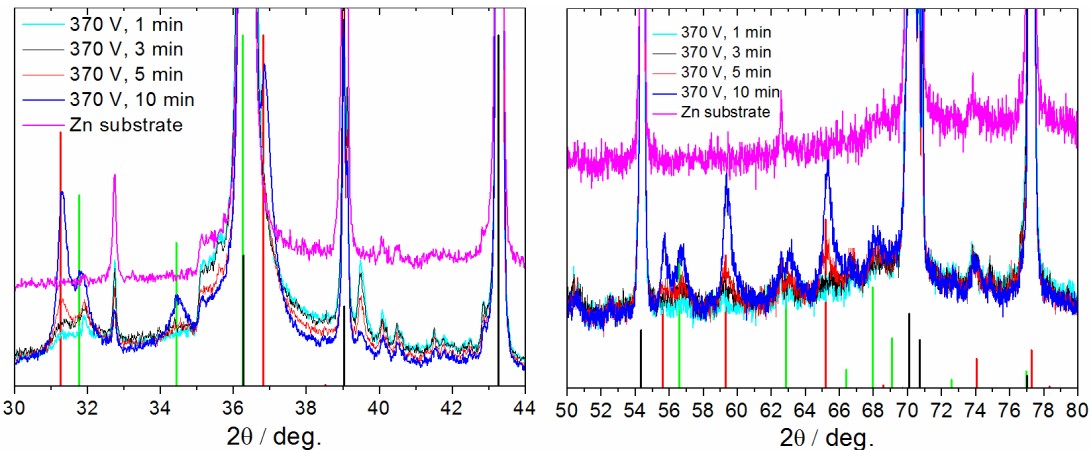

**Figure 9.** X-ray diffractograms of the pure Zn alloy substrate and PEO coatings prepared on the Zn alloy substrate at 370 V for different anodization time in an alkaline aluminate based electrolyte. Vertical lines correspond to the XRD peaks of ZnO (green; JCPDS: 36-1451), $ZnAl_2O_4$ (red; JCPDS: 82-1036) and metallic Zn (black; JCPDS: 04-0831) phases.

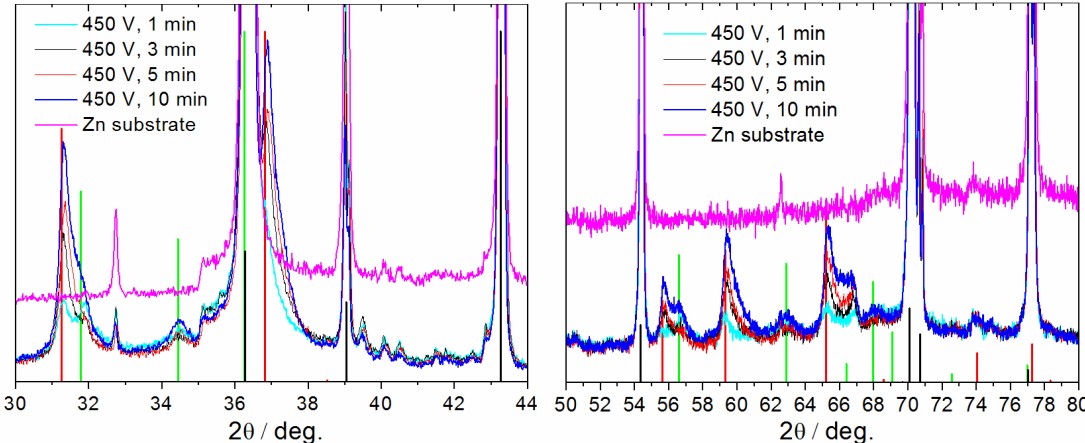

**Figure 10.** X-ray diffractograms of the pure Zn alloy substrate and PEO coatings prepared on the Zn alloy substrate at 450 V for different anodization time in an alkaline aluminate based electrolyte. Vertical lines correspond to the XRD peaks of ZnO (green; JCPDS: 36-1451), $ZnAl_2O_4$ (red; JCPDS: 82-1036) and metallic Zn (black; JCPDS: 04-0831) phases.

The appearance of the zinc aluminate phase in the PEO coating can be associated with the participation of aluminate anions, contained in the electrolyte, in various chemical and thermochemical processes that occur during PEO treatment. In particular, the aluminate anions react with water in alkaline solutions to produce aluminium tetrahydroxy anions (Reaction 1) that can be decomposed near the anode surface due to the pH and thermal conditions created in the local regions of electrolyte adjacent to the surface discharges (Reaction 2) [14,46]:

$$AlO_2^- + 2H_2O \rightarrow Al(OH)_4^- \tag{1}$$

$$2Al(OH)_4^- \rightarrow Al_2O_3 + 2OH^- + 3H_2O \tag{2}$$

Finally, owing to a very high temperature in the microplasma channels, $ZnAl_2O_4$ can be formed in accordance to the Reaction 3:

$$ZnO + Al_2O_3 \rightarrow ZnAl_2O_4 \tag{3}$$

### 3.4. Diffuse Reflectance Spectra of the PEO Coatings

To characterize the optical properties of the PEO coatings on Zn, which are important for different applications such as photocatalysis, the diffuse reflectance spectra (DRS) of the coatings formed at various stages of PEO process were analyzed. Figure A3 presents the DRS spectra of the coatings grown at 370 V and 450 V for different time. These spectra were recorded relative to the reference standard—$BaSO_4$ powder.

For analysis of DRS spectra, the Kubelka-Munk theory [47] generally is applied. According to this theory, the relative reflectance of the powder, $R_\infty$, can be converted into an equivalent absorption coefficient, $\alpha$, using the Kubelka-Munk function $F(R_\infty)$ calculated by the equation:

$$F(R_\infty) = (1 - R_\infty)/2R_\infty = \alpha/S, \tag{4}$$

where S is the coefficient of light scattering.

The calculated spectra of $F(R_\infty)$ for the PEO coatings are shown in Figure 11.

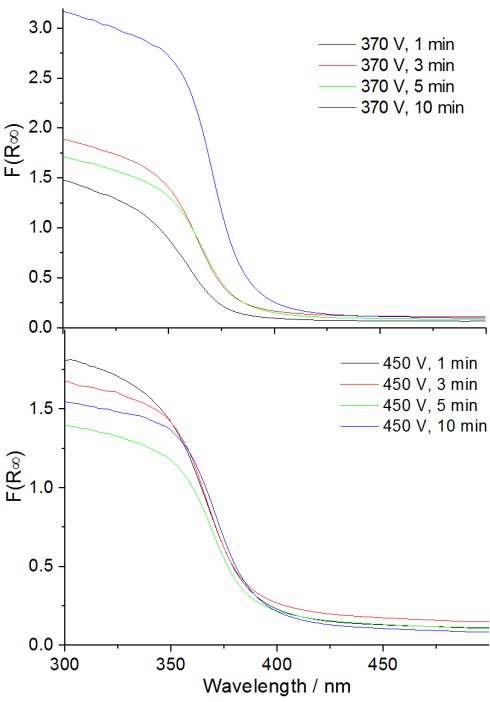

**Figure 11.** Absorption spectra of the PEO coatings prepared at 370 and 450 V for different anodization time in alkaline aluminate based electrolyte.

The energy dependence of the absorption coefficient $\alpha$ for semiconductors in the region near the absorption edge is given by the equation:

$$\alpha = A\,(h\nu)^{-1}\,(h\nu - E_g)^{1/n}, \tag{5}$$

where $E_g$ is the optical absorption edge energy, $\nu$ is the frequency of the incident photon, $h$ is the Planck constant, A is a constant, n = $^1/_2$ and n = 2 for allowed indirect and direct optical transitions, respectively.

Taking into account Equations (4) and (5), the energy intercept of a plot $(F(R_\infty)\,h\nu)^2$ as a function of $h\nu$ (Tauc plot) will give the band gap values for a direct allowed transition when the linear region of the plot is extrapolated to the zero ordinate. We used n = 2 in these calculations, because direct optical transitions are characteristic of both ZnO and ZnAl$_2$O$_4$. Using this method, the apparent band gaps for the PEO coatings are calculated (Figure A4) and listed in the Table 2. Since the studied coatings are highly defective, the values shown in Table 2 reflect both the changes in the bandgap and the appearance of defective states. The calculated values should be considered as apparent bandgap that does not necessarily reflect the fundamental absorption edge [48].

**Table 2.** Apparent band gap values calculated using DRS spectra of the PEO coatings prepared at 370 and 450 V for different anodization time.

| Time of the PEO Treatment, min | Applied Voltage, V | |
| --- | --- | --- |
| | 370 | 450 |
| 1 | 3.38 eV | 3.29 eV |
| 3 | 3.33 eV | 3.29 eV |
| 5 | 3.32 eV | 3.27 eV |
| 10 | 3.28 eV | 3.26 eV |

As can be seen from Table 2, the optical band gap of the PEO films is in the range from 3.38 to 3.26 eV, depending on the applied voltage and the PEO treatment time. These $E_g$ values are close to or slightly higher than the band gap of ZnO (about 3.2 eV [49]), but noticeably lower than the band gap reported for ZnAl$_2$O$_4$ (3.8 ÷ 4.2 eV [50–53]). According to the XRD data discussed above, the relative ZnAl$_2$O$_4$ content in the coating grows with increasing the applied voltage and treatment time. Therefore, the apparent $E_g$ of the composite coating is also expected to increase, but in reality it is reduced (see Table 2). The observed contradiction can be explained by the presence of a pronounced shoulder at the long-wavelength edge of the absorption spectrum of ZnAl$_2$O$_4$ powders, which was observed in the wavelength range of 320–400 nm [51,54]. This shoulder was tentatively related to electronic transitions between filled O 2p orbitals and empty 4s orbitals and would be representative of a defective structure [51]. The contribution from this shoulder to the total absorption in the wavelength range from 300 to 400 nm could be responsible for reduced values of the apparent $E_g$ for the PEO coatings with an enhanced content of ZnAl$_2$O$_4$.

### 3.5. Spectral Characterization of Microdischarge Emission during PEO Process on Zn Electrodes in Aluminate-Based Electrolyte

Measurement and analysis of radiation spectra from microdischarges using optical emission spectroscopy (OES) provides a valuable diagnostic tool for studying the PEO process [55–57]. As mentioned above, the PEO process on Zn alloy begins from "soft sparking" mode when small microdischarges with white color appear on the electrode surface. The spectrum of these microdischarges observed during the first 5 min at 370 V does not demonstrate specific emission lines and only a broad halo in the visible region (450–850 nm) is registered. The appearance of small white sparks can be explained via the presence of free electrons in the plasma interacting with solid and/or liquid compounds and bremsstrahlung radiation [56]. When microdischarges become more intense and their

color changes from white to yellow, the spectrum of emission is changed. Typical spectrum corresponding to this stage is shown in Figure 12. The strong lines appear in the spectra and can be attributed to Na I (588.99 nm, 589.59 nm) and K I (766.57 nm, 769.94 nm) lines originating from the electrolytes.

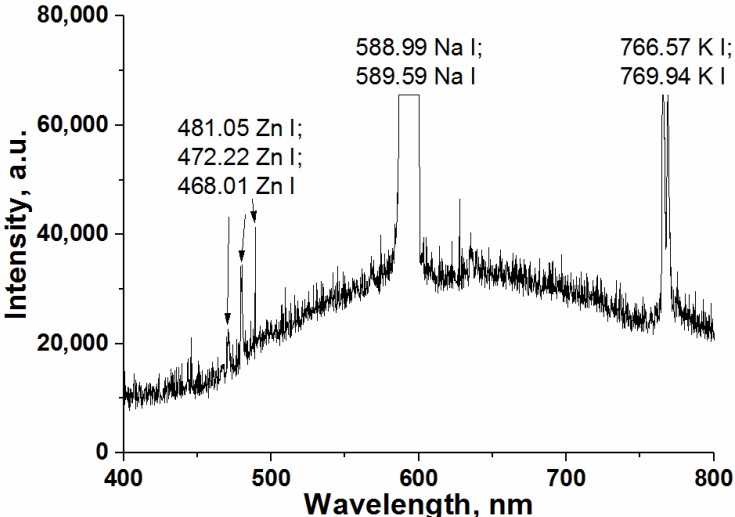

**Figure 12.** Optical emission spectrum recorded during PEO of Zn alloy in aluminate electrolyte.

In order to better identify less intensive species originated from zinc anode and electrolytes, the detailed optical emission spectra have been analyzed in range 370–420 nm and 450–500 nm, respectively (Figures 13 and 14). Three emission lines of Zn I at 468.01 nm, 472.22 nm and 481.05 nm are observed, which originate from the zinc anode (Figure 13). With increasing the applied voltage up to 450 V when the sparks become significantly larger, additional lines appear in the optical emission spectra. These lines arise from components of aluminate electrolyte and can be assigned to Al I at 394.40 nm, Al I at 396.15 nm, O II at 404.49 nm, H I at 656.28 nm (Figure 14).

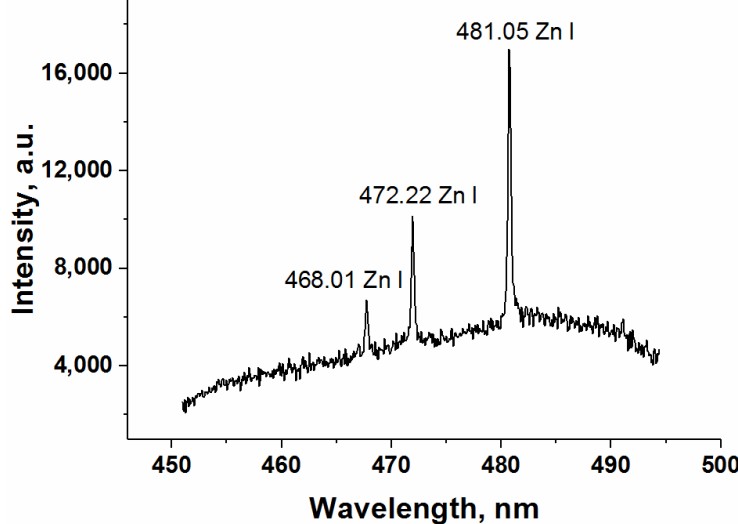

**Figure 13.** Optical emission spectrum in the range from 450 to 500 nm recorded during PEO of Zn alloy at 370 V.

According to Ref. [56], three discharge models have been developed for the interpretation of the discharge appearance during PEO process: metal-oxide interface discharge (type B), oxide-electrolyte interface discharge within the coating upper layer (type A) and at the coating top layer (type C). Since the melting point of Zn (419.6 °C) is relatively low,

the type B of microdischarge appearance may be present during PEO process. At the same time, oxygen, potassium, sodium and aluminium lines in the optical emission spectra are related to processes of type A and type C. The XRD and EDS data of the obtained PEO coatings (presented above) coincide with the OES results.

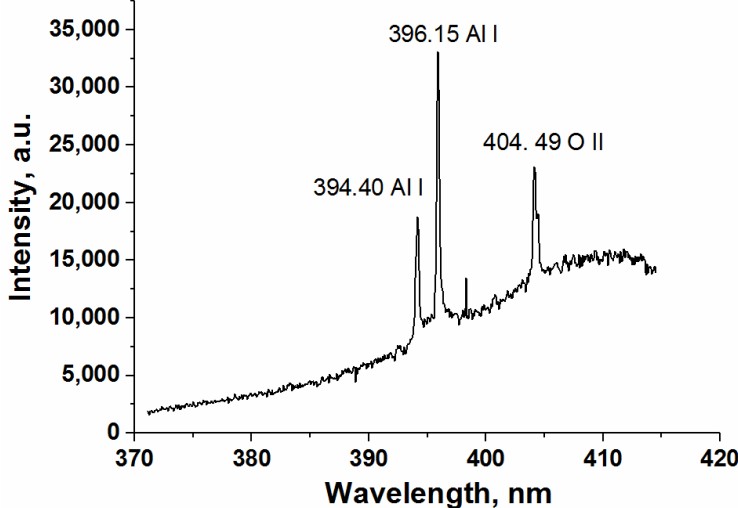

**Figure 14.** Optical emission spectrum in the range from 370 to 415 nm recorded during PEO of Zn alloy at 450 V.

*3.6. Photoluminescence Properties of the PEO Coatings*

Interest in studying the photoluminescence (PL) properties of zinc oxide based materials is associated with the perspectives for the practical application of such materials in various fields [58,59]. In particular, the use of luminescence of ZnO materials with a large specific surface as a probe of gas adsorption has been highlighted in some previous works [60,61].

Typical PL excitation spectra (monitored at 570 nm) and corresponding emission spectra (excited at 340 nm) of the PEO coatings prepared at 370 and 450 V for different anodization time are shown in Figures 15 and 16, respectively.

There are two bands in the excitation spectra of the PEO coatings prepared at 370 V for 1–5 min and at 450 V for 1 min (Figure 15). Maximum of the first band located at $335 \div 360$ nm is shifted to shorter wavelengths with decreasing the treatment time. Position of the other band located at ~275 nm slightly depends on the processing parameters. This band practically disappears with an increase in processing time up to 10 min at 370 V and up to 3 min or more at 450 V. Instead of this band, an exponential rise in the PL intensity appears, starting from 300 nm towards shorter wavelengths (Figure 15). The observed evolution of the PL excitation spectra with a change in the PEO processing parameters can be related to a change in the phase composition of the PEO coatings. As it is shown by XRD analysis (Section 3.3), the relative $ZnAl_2O_4$ content in the coatings grows with increasing the treatment time and applied voltage. Thus, the PL excitation spectrum with exponential growth in the short-wavelength region can be attributed to $ZnAl_2O_4$ phase. Actually, this semiconductor material has a wider band gap than ZnO, and similar PL excitation spectra were reported previously for $ZnAl_2O_4$ powders [25,28,62].

It should be noted that the shape of excitation and emission spectra is changed with changing the emission and excitation wavelengths. Figure 17 shows the evolution of the PL spectra excited at 275 nm, depending on the parameters of the PEO process.

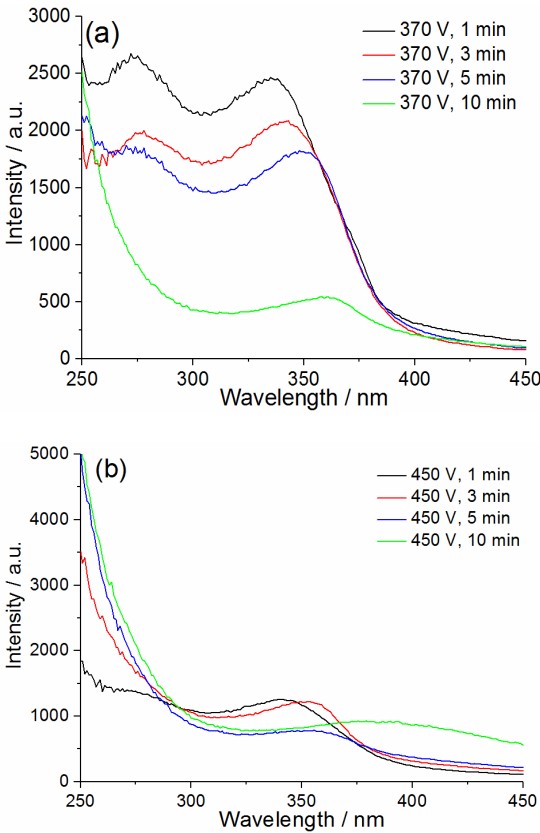

**Figure 15.** PL excitation spectra of the PEO coatings prepared at 370 V (**a**) and 450 V (**b**) for different anodization time. PL emission was monitored at 570 nm.

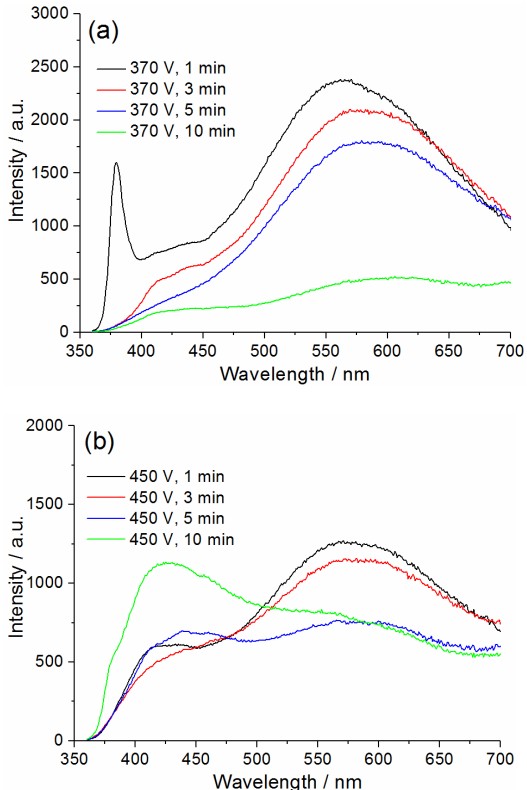

**Figure 16.** PL emission spectra of the PEO coatings prepared at 370 V (**a**) and 450 V (**b**) for different anodization time. PL was excited at 340 nm.

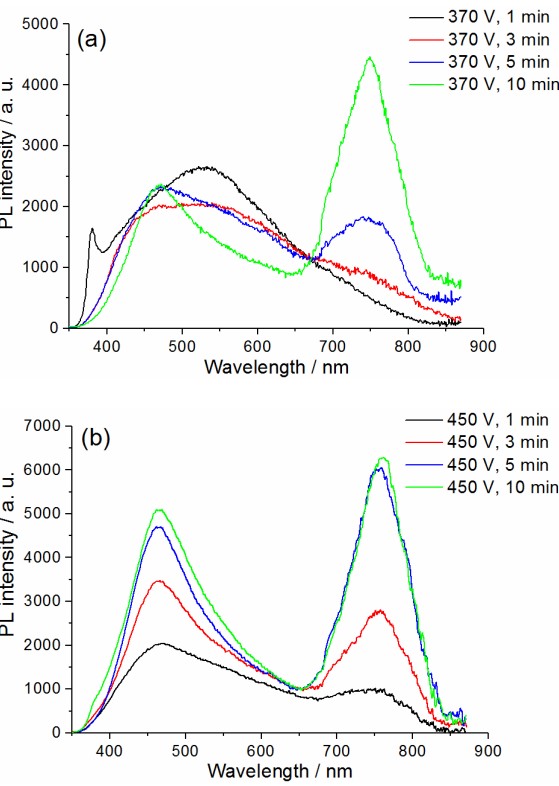

**Figure 17.** PL emission spectra of the PEO coatings prepared at 370 V (**a**) and 450 V (**b**) for different anodization time. PL was excited at 275 nm.

PL emission spectra of the PEO coatings are composed of several bands located mainly in the visible region (Figures 16 and 17). It should be noted that the total photoluminescence of the coatings is a sum of the PL originating from ZnO and $ZnAl_2O_4$. Based on the XRD data, it is logical to assume that for the coatings grown at 370 V for 1 min, the main contribution to the PL will be made by ZnO. Actually, only these coatings are characterized by a sharp near-ultraviolet PL band centered at ~380 nm, which can be assigned to radiative recombination of free excitons in ZnO [63]. In addition to this short-wavelength band, other wide overlapping bands are observed in the visible region. ZnO commonly demonstrates PL properties in the visible spectral range due to the different intrinsic or extrinsic defects. The origin of this visible PL (green, yellow, orange, and red emission) is still highly controversial. Generally, the green PL is typically associated with oxygen deficiency (e.g., excess $Zn^{2+}$ ions or double ionized oxygen vacancies), and the yellow/orange PL is related to excess oxygen (e.g., oxygen interstitial defects) [64–67]. The wide visible PL band in the 400–800 nm range observed for the PEO coatings grown at 370 V for 1 min indicates that simultaneously several types of defects can contribute to the radiative recombination (Figures 16 and 17). With increasing anodization time and applied voltage in the PL spectra of coatings, this wide PL band shifts to shorter wavelengths and its width decreases. This effect is especially pronounced in the case of PL excited by shorter wavelength light (275 nm) (Figure 17). The emerging PL band peaked at ~460 nm can be assigned to $ZnAl_2O_4$. Similar PL emission spectra were previously reported for pure $ZnAl_2O_4$ powders, and the observed blue emission was ascribed to intra-band-gap defects, such as oxygen vacancies, in $ZnAl_2O_4$ crystals [68]. At the same time, near infrared emission around 760 nm appears under 275 nm excitation. This band grows with increasing the anodization time and applied voltage, which may be due to an increase of $ZnAl_2O_4$ content in the coating. To confirm this observation, the $ZnAl_2O_4$ powder has been obtained. The synthesis of $ZnAl_2O_4$ powder is described in Appendix A. The $ZnAl_2O_4$ spinel structure of prepared powder has been confirmed by XRD (Figure A5). We found two bands (one weak band centered at 570 nm and another strong band at 765 nm) in the emission spectra of the

prepared $ZnAl_2O_4$ powder annealed at 800 °C. Increasing the annealing temperature to 1000 °C leads to a shift of the emission to lower wavelengths and a significant decrease in emission in the near-infrared range (Figure 18).

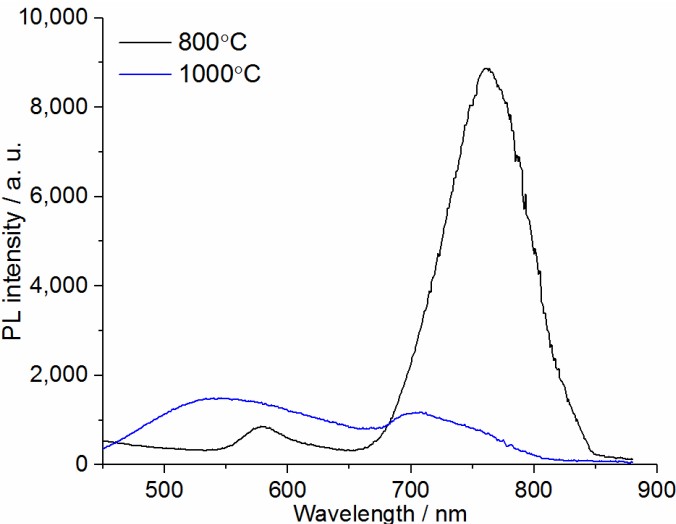

**Figure 18.** PL emission spectra of the $ZnAl_2O_4$ powders annealed at 800 °C and 1000 °C for 5 h. PL was excited at 275 nm.

According to Ref. [62], such near-infrared emission is strongly dependent on annealing temperature of $ZnAl_2O_4$ and originates from structural defects due to formation of oxygen vacancies. This observation allows concluding that $ZnAl_2O_4$ synthesized during PEO is poorly crystallized and have high level of structural defects. Thus, varying the processing parameters of PEO treatment of Zn alloy can be effective tool for tuning luminescent properties of $ZO/ZnAl_2O_4$ coatings.

## 4. Conclusions

Plasma electrolytic oxidation of zinc alloy Z1 has been studied in an alkaline aluminate-based electrolyte. SEM study showed that the prepared PEO coatings have fused surfaces with randomly distributed microcraters. According to EDS analysis, the main elemental components of the obtained coatings are Zn, Al and O. XRD results demonstrated that the coatings are crystallized and composed of ZnO (wurtzite) and $ZnAl_2O_4$ (spinel) phases. The phase composition of the coatings is changed with increasing the applied voltage and anodization time: the content of ZnO phase decreases and the proportion of $ZnAl_2O_4$ increases. The excited atoms of H, O, Na, K, Zn and Al, which were identified in optical emission spectra of microdischarges, originate from zinc substrate or electrolyte components. A set of characterization techniques (XRD, EDS-mapping and OES) revealed that the formation of a zinc oxide layer precedes the incorporation of aluminium in the form of zinc aluminate into the coatings as energy of plasma discharges increases. The apparent optical band gap of the PEO composite coatings is estimated from DRS spectra and is in the range from 3.38 to 3.26 eV depending on the applied voltage and PEO treatment time. It has been found that the band gap is reducing during the increase of $ZnAl_2O_4$ content in the PEO coatings. PL excitation and emission spectra of the PEO coatings prepared on the Zn alloy demonstrate a complex evolution of the shape with changing the excitation and emission wavelengths. PL of coatings prepared at 370 V for short times is basically associated to ZnO, whereas increasing the applied voltage and anodization time leads to the appearance of PL bands (peaked at 460 and 760 nm) characteristic of $ZnAl_2O_4$. The appearance of PL band in near-IR region is associated with the defective structure of $ZnAl_2O_4$. The luminescent properties of PEO coatings can be flexibly tuned by PEO processing parameters due to the combination of $ZnAl_2O_4$ and ZnO. Thus, the regulation of the $ZnAl_2O_4$ content and struc-

ture revealed the possibility for the creation of new optical devices and electroluminescence displays with desirable functionalities.

**Author Contributions:** Conceptualization, S.P. and M.S.; methodology, R.V. and S.P.; software, N.B.; validation, S.P. and S.S.; formal analysis, H.M., M.S. and S.P.; investigation, H.M., S.S., S.K. and S.P.; resources, S.P. and R.V.; data curation, H.M. and N.B.; writing—original draft preparation, H.M. and S.P.; writing—review and editing, H.M., R.V., N.B., S.K. and S.P.; visualization, N.B.; supervision, S.P. and R.V.; project administration, R.V.; funding acquisition, R.V. All authors have read and agreed to the published version of the manuscript.

**Funding:** This work was partially supported by FUNCOAT project ("Development and design of novel multiFUNctional PEO COATings") in frame of H2020-MSCA-RISE-2018, (Grant Agreement No. 823942), the Belarusian Republican Foundation for Fundamental Research (Grant No. X21M-073) and the State Program for Scientific Research of Belarus "Chemical processes, reagents and technologies, bioregulators and bioorganic chemistry" (Project No. 2.1.04.02).

**Institutional Review Board Statement:** Not applicable.

**Informed Consent Statement:** Not applicable.

**Data Availability Statement:** Data available in a publicly accessible repository.

**Acknowledgments:** The authors thank Anastasiya Tabolich and the Center for Analytical and Spectral Measurements of the B.I. Stepanov Institute of Physics for measuring photoluminescence spectra.

**Conflicts of Interest:** The authors declare no conflict of interest.

## Appendix A

### *Appendix A.1. Adhesion/Cohesion Behavior*

The adhesion/cohesion of the PEO coatings prepared at 370 V and 450 V has been evaluated by pull-off tests using ONIKS 1.AP adhesion tester (Interpribor, Chelyabinsk, Russia).

Adhesion of $ZnO/ZnAl_2O_4$ coatings on Zn alloy prepared at 450 V and 370 V was approximately the same ($0.95 \pm 0.21$ MPa). The slightly greater adhesion ($1.3 \pm 0.11$ MPa) was achieved for PEO coatings prepared at 370 V for 1 and 3 min. It can be related to the increase of defects, internal stresses and cracks in the PEO coatings with increasing the treatment time. However, some part of PEO coating remains on the substrate after pull-off test due to formation of microstresses and cracks within the PEO layer. The obtained results are in a good agreement with the data presented in [24].

### *Appendix A.2. Wear Behavior*

The dry sliding wear behaviour of the PEO coatings was assessed with an oscillating ball-on-disc tribometer (Tribotec AB, Clichy-France, France), with an AISI 52100 steel ball of 6 mm diameter as the static friction counterpart (IHSD-Klarmann, Bamberg, Germany). The wear tests were performed at ambient conditions ($25 \pm 2$ °C and 36–44% relative humidity) with load of 1 N and an oscillating amplitude of 10 mm with a sliding velocity of 5 mm·s$^{-1}$. The test was terminated after a total sliding distance of 12 m. Laser scanning confocal microscope (LSM 800, ZEISS, Jena, Germany) was used for analysis of the wear tracks after the test. ConfoMap$^{ST}$ software (ZEISS, Jena, Germany) (version 1.0) was used for subsequent data treatment and analysis.

Figure A1 demonstrates the comparison of the coefficient of friction for the PEO coatings prepared at 370 and 450 V. The curves initially increase sharply and then equalizes to the average value of the coefficient of friction. The latter was approximately the same during 20 min of the measurement. However, the coatings prepared at 370 V after 20 min of the abrasion show the decrease of the friction coefficient. SEM images (Figure A2) of the wear tracks evidence totally removing the PEO coatings prepared at 370 V during the wear test. At the same time, the coatings obtained at 450 V do not fail during the test and the Zn substrate is not reached. The trend emerged from the wear measurement evidence that the regime of sparking contributes to the formation of more wear-resistant

PEO coatings. Thus, the coatings formed at "microarcs" mode are characterized by thick and well-crystallized PEO layers which demonstrate higher wear resistance in comparison with coatings prepared at "soft sparking" or/and "sparking" modes.

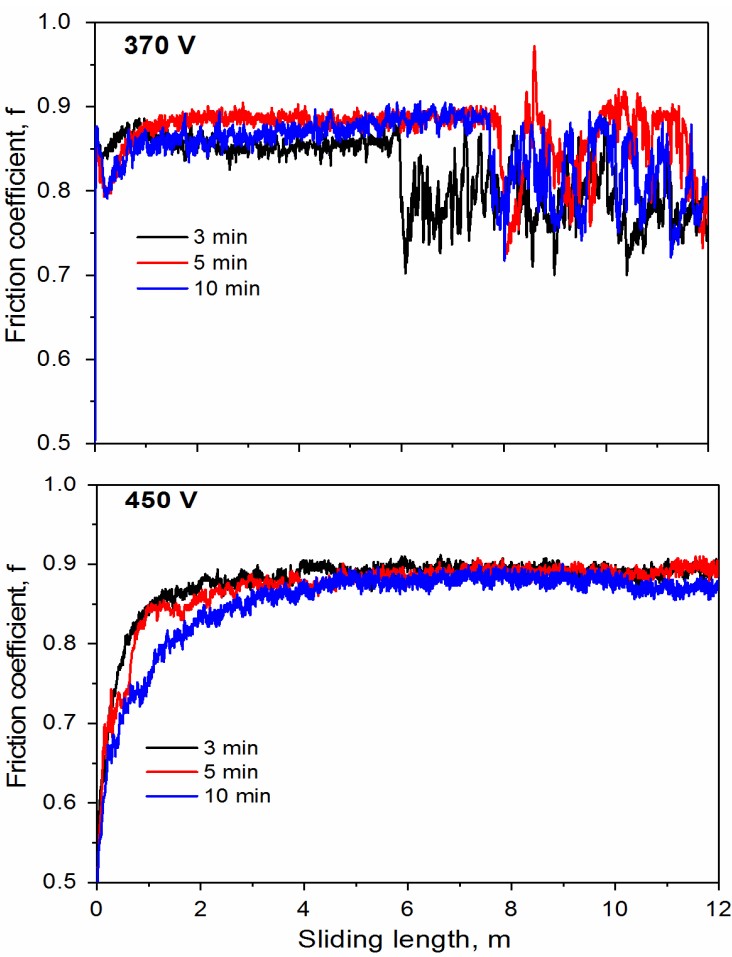

**Figure A1.** Comparison of the coefficient of friction for the PEO coatings prepared at 370 and 450 V during 3, 5 and 10 min at load of 1 N.

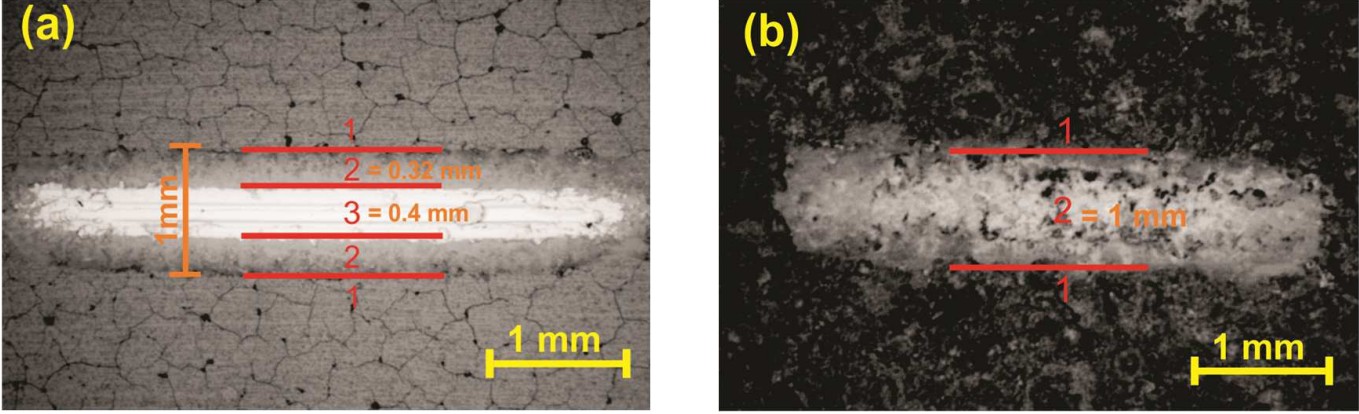

**Figure A2.** SEM images of the surface of PEO coatings prepared at 370 V (**a**) and 450 V (**b**) for 10 min after the wear tests.

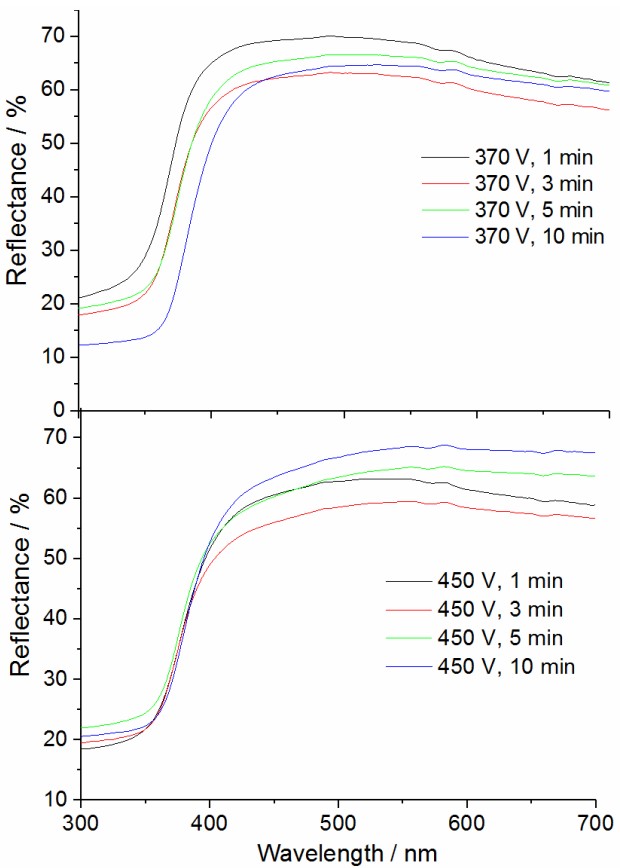

**Figure A3.** Diffuse reflectance spectra of the PEO coatings prepared at 370 and 450 V for different anodization time in alkaline aluminate based electrolyte.

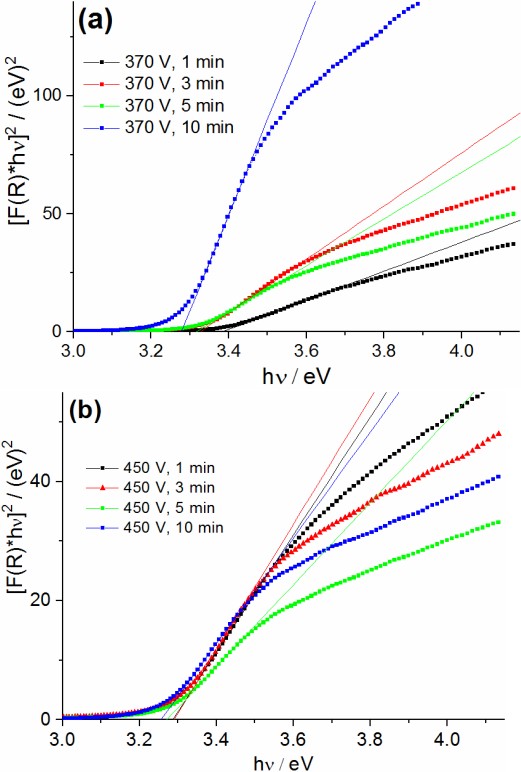

**Figure A4.** Tauc plots of the PEO coatings prepared at 370V (**a**) and 450V (**b**) for different anodization time.

*Appendix A.3. The Synthesis of ZnAl₂O₄ Powder*

ZnAl$_2$O$_4$ powder was synthesized by the co-precipitation method as described below followed by thermal treatment at 800 and 1000 °C. 500 mL of 0.02 M NaAlO$_2$ was mixed with 500 mL of 0.01 M Zn(NO$_3$)$_2$ under vigorous stirring. The obtained precipitate was separated from the solution and washed by centrifugation. Then, the powder was annealed at different temperatures (at 800 and 1000 °C) for 5 h. XRD patterns of prepared powders are presented in Figure A5.

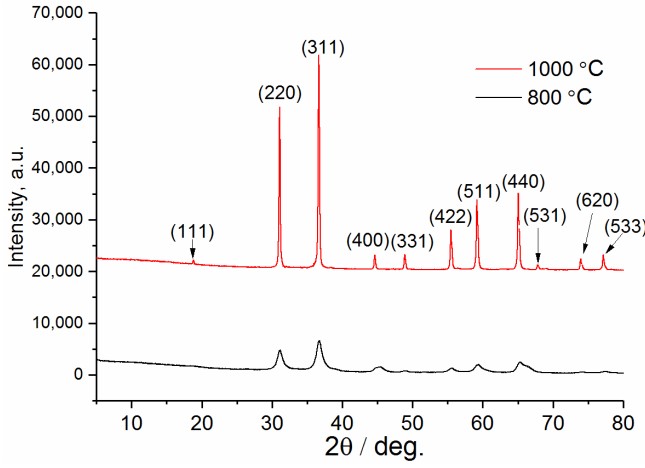

**Figure A5.** XRD patterns of ZnAl$_2$O$_4$ annealed at 800 and 1000 °C.

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
