# Peer review of "Photoluminescent Coatings on Zinc Alloy Prepared by Plasma Electrolytic Oxidation in Aluminate Electrolyte"

_coatings, doi:10.3390/coatings13050848_

Round 1
Reviewer 1 Report
Referee report on “Photoluminescent coatings on zinc alloy Z1 prepared by plasma electrolytic oxidation in aluminate electrolyte”
This is a rather interesting and good paper that certainly can be recommended for publication, but clarifying and detailing some parts of the text.
1. Line 77. “The goal of the present work is the preparation …” Preparation for what?
More recent applications
Probably, it is important to note that these and similar materials have many important applications as optical materials, or materials for nuclear/fusion technology, such as diagnostic/detector components and materials for shielding etc. It would be useful to mention a few recent applications of ZnAl2O4.
https://scholar.google.lv/scholar?q=ZnAl2O4&hl=en&as_sdt=0%2C5&as_ylo=2021&as_yhi=
2. Paragraph 3.4 and Fig.A2.
The analysis of the data in Figure A2 needs serious discussion and justification. See the latest article by the Editors of “Optical Materials” journal:
Brik, M. G., Srivastava, A. M. (2022). A few common misconceptions in the interpretation of experimental spectroscopic data. Optical Materials, 127, 112276.
This is all the more important when the authors later in the article confirm that their films are defective. The legitimacy and accuracy of the drawn lines and their belonging to the edge of fundamental absorption, and not to defective levels at the absorption edge, requires a more serious definition.
3. Line 318. “… various fields [48]”. Is it still relevant? Are there newer reviews?
4. Lines 319-320. Are there also newer results?
5. Lines 334-336. “Thus, the PL excitation spectrum with exponential growth in the short-wavelength region can be attributed to ZnAl2O4 phase.” Why? Actually, it follows from the given data that the Eg is greater than 5 eV. (Fig.15b)
6. Do the films contain fine inclusions of Al2O3? These inclusions may not be visible in diffraction, but are easily visible in photoluminescence.
7. Line 343-363. The approach commonly used for ZnO is considered here. At the same time, it is important to note that the behavior of ZnAl2O4 should be closer to that of MgAl2O4 and Al2O3. This is clearly evidenced especially by the excitation spectra. Moreover, the luminescence spectra are very similar to MgAl2O4 and Al2O3.
G Prieditis et al 2019 IOP Conf. Ser.: Mater. Sci. Eng. 503 012021
Antuzevics, A., 2023 Optical Materials, 135, p.113250.
8. It is known that the MgAl2O4 spinel can be of different stoichiometry. How was the stoichiometry checked in this work ZnAl2O4?
9. It would be useful if the data in Figures 9 and 10 were supplemented with Raman measurements.
10. It would be interesting to know why the decomposition of the luminescence spectra into Gaussians was not done?
In general, the manuscript is interesting and can be recommended for publication after constructive reflection on the above comments.
Author Response
We thank the reviewer for the questions and recommendations. Please, find our response in the attachment.

Reviewer 2 Report
-I dont think it is proper to use both 'zinc alloy' and 'Z1' while describing. I recommend to use only zinc alloy substrate or Z1 after giving the desciribng in bracket.
-Please do not use parentheses in keywords.
-The statement of novelty of this work is not clear. Please give the novelty in the last paragraph in detail. What exactly is the gap filled in the literature?
-Can you provide EDS mapping image including EDS spectrums for coating morphology?
-Have pull-off tests been done? It is critical for adhesion and cohesion and recommended.
-A scientific approach is essential on which all the results obtained are based. For this, discussion should be developed for each result obtained.
-Please write the obtained results in the light of concise and scientific point of view, for conclusion section.
-Did the authors study the wear and corrosion properties of the resulting coatings?
Author Response

(The authors gave the same response as above.)

Reviewer 3 Report
The manuscript entitled " Photoluminescent coatings on zinc alloy Z1 prepared by plasma electrolytic oxidation in aluminate electrolyte " mainly focuses on the characterization and investigation of the thick ZnO/ZnAl2O4 coatings on zinc alloy Z1 substrate prepared by plasma electrolytic oxidation technique.
The material presented in this manuscript corresponds to the logical sequence characteristic of a scientific article. Nevertheless, I have a few critical observations that radically change the approach to the information presented in this manuscript.
The materials used in this work are of low purity, so the obtained coatings are inhomogeneous with a high amount of impurities. The XRD analysis confirms this conclusion.
The presentation of the XRD diffractograms (Figs. 9 and 10) is very unusual. The peaks in them are not indexed and the crystalline phases are not identified.
The results of the EDS analysis are not accurate, since the used holder is made from aluminium. The amount of Al and oxygen values in Table 1 are questionable.
The angle for determining the thickness of the coating is not equal to 90o degrees, so the obtained result (Figs. 6 and 7) is very inaccurate. In addition, there is no general picture of the coating thickness, and the measurement locations were selected according to the desired need.
The authors do not justify the applied voltage modes.
In summary, it is obvious that the chosen synthesis method is not suitable for the preparation of corresponding coatings and the optical properties do not have any logical sequence of changes.
Author Response

(The authors gave the same response as above.)

Round 2
Reviewer 1 Report
The authors have successfully improved their original manuscript, which now can be
recommended for publication.
Reviewer 2 Report
Thanks the authors for their proper revisions mentioned by me in previous version.
Reviewer 3 Report
No comments.